# The immediate treatment outcomes and cost estimate for managing clinical measles in children admitted at Mulago Hospital: A retrospective cohort study

**Barbara Namugga**[1]*, **Ombeva Malande**[1], **Jonathan Kitonsa**[2], **Leonard Manirakiza**[3], **Cecily Banura**[4], **Ezekiel Mupere**[1]

**1** Kampala Uganda Department of Paediatrics, Makerere University College of Health Sciences, Kampala, Uganda, **2** Medical Research Council/Uganda Virus Research Institute and London School of Hygiene and Tropical Medicine, London, England, **3** Uganda National Bureau of Standards, Kampala, Uganda, **4** Kampala Uganda Department of Medical Epidemiology and Biostatistics, Makerere University College of Health Sciences, Kampala, Uganda

* barbaranamugga28@gmail.com

**Data Availability Statement:** Data cannot be shared publicly as it contains confidential and potentially identifying patient information, and

## Abstract

Over the recent years, the Ministry of Health in Uganda has reported multiple measles outbreaks in various districts despite the availability of a safe cost effective vaccine. Measles, especially among the unvaccinated can lead to serious complications including death while its management heavily burdens the family and health care system. This study aims to determine the immediate treatment outcomes and estimate the cost of treating a measles case. A retrospective cohort study using records review was conducted among children 0–12 years admitted at Mulago hospital throughout 2018. Demographics, complications, vaccination status, discharge status, duration of hospital stay, type of treatment, supplies and investigations done were abstracted from the patient charts. Treatment costs were obtained from the hospital pharmacy price list while the unit cost of utilities, human resource, food and security were obtained from the hospital accounts department. Patients' characteristics were summarized descriptively. Cost information, was reported as mean with standard deviation (SD) and range, and was stratified and presented as direct health care (blood test, radiology and treatment) and direct non health care costs. Among 267 reviewed patient charts, the median age was 1.0 ((IQR 0.75–2) years. 63patients (24%) were immunised, 79 (29%) were not immunized, Median length of hospital stay was 4.0 days (IQR 3.0–7.0) with majority (n = 207, 77%) staying < 7 days. 30 patients (11%) died with mortality highest among the unimmunised (n = 13, 44%) and severe pneumonia (39.5%) was the commonest complication. 114.5 USD was estimated to treat a child with measles. Human resource (79.33USD, SD 4.63) and treatment costs (21.98USD, SD 22.77) were the largest expenses. Complications are common in majority of fatal measles cases and these carry a high cost to the healthcare system.

because they are the property of the Mulago National Referral Hospital. Request for data can be made to the Chairman School of Medicine Makerere University College of Health Sciences Institution Review Board (contact via: ponsiano. ocama@gmail.com, + 256772421190) for researchers who meet the criteria for access to confidential data.

**Funding:** This work was supported by the University of Minnesota to Barbara Namugga through CB. However, the funders had no role in study design, data collection and analysis, decision to publish, or preparation of the manuscript. The authors did not receive any salary from the funders.

**Competing interests:** The authors have declared that no competing interests exist.

## Introduction

Measles is a highly contagious viral disease which is largely preventable through vaccination [1]. Globally, measles accounts for over 140,000 deaths annually, mostly among children under five [1] and is still common in many developing countries–particularly in parts of Africa and Asia. The overwhelming majority (more than 95%) of measles deaths occur in countries with low per capita income and weak health infrastructure [1–3].

In Uganda, the Ministry of Health continues to register measles outbreaks almost every year despite the availability of free effective vaccines in public health facilities, with Wakiso and Kampala districts amongst those most commonly affected [4]. Outbreaks have also been reported in several other districts, including a recent one in Nakaseke [5], and a more widely spread one in 2018 involving 26 districts [6]. As a result, measles is one of the leading causes of death among children under 5 years in the country, and contributes 4% to under 5 mortality [7].

To prevent measles outbreaks, WHO had set an ambitious but achievable target of vaccinating $\geq$ 95% of the susceptible children by 2020, which would help create herd immunity [8]. Findings from the Uganda Demographic and Health survey (2016) showed variation between districts in measles vaccine coverage, which was less than optimum. Coverage ranged between 60% in Wakiso and 82.8% in Kampala with a national average of 80% [9]. Low levels of immunization coverage translate to low immunity at community level, which ultimately may lead to an outbreak of measles if the virus is introduced.

Besides morbidity and mortality, costs related to treating a child with measles pose a big burden to the family and the already financially stretched health care delivery system, and are increased when the measles is complicated with conditions such as pneumonia, encephalitis, malnutrition, diarrhoea, blindness, croup, otitis media among others. While the cost of immunizing a child is estimated at only $0.86 [$0.42 being cost for vaccine and injection material and $0.44 for operational costs] [10], the cost incurred to the health care delivery system in managing measles in Uganda has not been well described. This study therefore sought to estimate the average cost to the health care delivery system in managing a child with measles in 2018 when the largest measles outbreak occurred in Uganda. Knowledge of this would inform policy related to promotion of immunisation and planning for hospital services.

## Methods

### Study design and setting

A retrospective cohort study through records review was conducted at Mulago National Referral Hospital (MNRH), which is the teaching Hospital of Makerere University College of Health Sciences. This Public health facility is owned by the Uganda Ministry of Health, and is located in the central division of Kampala district. MNRH serves about 115,000 in and out patients including 30,000 children per year with a bed capacity of 1790.

In Uganda, Measles vaccine is given at 9 months, and is freely available at all government health facilities as well as some private facilities. In times of epidemics, the vaccine is also often administered widely through mass vaccination campaigns.

### Study procedure

Data collection was carried out in a period of six months between January 2019 and June 2019. A list of the measles admissions from 1$^{st}$ January 2018 to 31$^{st}$ December 2018 with inpatient numbers was made from the Health Management Information System register to determine the total number of measles admissions at MNRH by the principal investigator with the help

of the hospital records officer. The available charts of children 0–12 years were retrieved by the records officer and reviewed for eligibility of data by the principal investigator. All charts with a clinical diagnosis of measles were included, while charts with illegible handwriting, undocumented inpatient number, age, and sex were excluded. The clinical diagnosis of measles was confirmed if the child's clinical features included fever, maculopapular rash with either cough, coryza, or conjunctivitis [11]. Sample size was determined based on the formula by Yamane 1967 [12,13];

$$n_r = \frac{N}{1 + N(e)^2}$$

where N is the sampling frame estimated at 812 from the Mulago Hospital medical admission records (HMIS inpatient register) and e is the allowable error, taken as 5%. The sample size used 267 patient records.

Charts that passed these criteria were arranged and grouped by date of admission starting with January 1st 2018. Systematic sampling technique, was used to pick the charts using a skip interval (k = 2), implying that every second chart was chosen until the required sample size was achieved. The first chart was selected randomly.

Information abstracted from the patient's charts included: Demographics (age, sex), complications, co-morbidities, vaccination status, vital status (alive or dead) at discharge and date of admission and discharge/death. The type of treatment, dosage and duration of use, supplies and investigations were also obtained from the patient charts and their unit costs were obtained from the hospital pharmacy price list.

Costs were classified as direct health care (treatment costs, i.e. drugs and supplies, and investigations, i.e. blood tests and radiology tests) and direct non-health care/overhead costs (security, salaries, water, electricity, food). The direct non-health care costs were obtained from the hospital planning unit/accounts department and costed per day per patient. Monthly salaries of staff who work on wards where measles patients are admitted were obtained from the hospital accounts department and daily salaries were calculated. This was then divided by the average number of patients seen per day to get the average cost. Unit costs were obtained using the price list from the hospital. Market prices were used for those items that were not in the price list. Total costs per patient were calculated and average costs determined. All costs incurred were assumed to have been received by the patient since they would be required in the management of measles.

## Data management and analysis

Each data abstraction form (S1 File) was assigned a study number, which was used alongside the inpatient numbers to enter data. These were checked against the patient's record for completeness and accuracy. The abstracted data was stored in a safe place under lock and key. Data was entered into an electronic database using Epidata version 3.1 with in-built quality control checks. The final data was then backed up, stored on a password-protected computer, and exported to STATA version 13 for analysis.

Continuous variables were summarised using medians with interquartile ranges, and means with standard deviations. The data was examined for the normality statistical assumption. Analysis of variance (ANOVA) were used to compare the average direct costs across patient demographic and clinical characteristics, and a P-value of $< 0.05$ was considered statistically significant.

### Ethics approval and consent

Ethical approval was sought and obtained from School of Medicine Research and Ethics Committee (SOMREC) (REF 2019–009) and The Mulago Hospital Institutional Review Board. A waiver of consent was sought and obtained from SOMREC to use the patient charts. All the information was kept confidential.

## Results

A total of 570 patients were admitted between 1st January and 31st December 2018 with a clinical diagnosis of measles and a total of 553 (97%) charts were retrieved. We excluded 17 charts: Seven of them had missing documentation of sex and age and 10 had no clear definition of clinical measles. This information is summarised in Fig 1.

### Demographic and clinical characteristics

After sampling, 267 patients' charts were chosen, of which 135 (51%) were for females and 132 (49%) for males. The median age was 1.0 year (IQR 0.75–2.0) with majority aged < 1 year (n = 169, 63%). The median length of hospital stay was 4.0 days (IQR 3.0–7.0). Thirty (11%) died during admission whereas 237 (89%) were discharged alive as presented in Table 1.

### Immediate treatment outcomes

**Complications developed during hospital stay.**   During the period of admission, 185 (69.3%) patients had multiple complications and these included severe pneumonia (39.5%), gastroenteritis (24.0%), conjunctivitis (18.0%), malnutrition (6.4%), encephalitis (2.2%), among others as illustrated in the Fig 2. Majority 29.2% of patients who developed complications were not immunised. The immunisation status of these 185 children is shared as S1 Table.

**Mortality of patients with measles and their characteristics.**   A total of 30 (11.3%) patients died during hospital stay. Out of these, 17 (57%) were male. 21 (70%) of these were aged between 0–1 years and 13 (44%) were not immunized. Majority 26 (86%) had spent < 7 days in the hospital as summarised in S2 Table.

**Complications among the 30 patients who died during admission.**   The common complications among the patients who died included severe pneumonia (28%), gastroenteritis (20%), conjunctivitis (13%), malnutrition (9%) among others as illustrated in Fig 3.

**Characteristics of the 60 patients who stayed longer than 7 days.**   Sixty (23%) patients were admitted for longer than 7 days. Of these, 46 (77%), were less than one year and 36 (60%) were male as shown in S3 Table.

### Costs of managing a child with measles

**Direct health care costs (USD).**   Direct health care costs were divided into 3 categories i.e. blood tests, radiology and treatment (drugs and supplies) costs. The average cost of blood test per patient was 11.73USD (SD = 12.10); for radiology, the average cost per patient was 3.48USD (SD = 4.83); and 21.98USD (SD = 22.77) for treatment. The overall average cost per patient was 33.13 USD (SD = 30.45).

**Direct non health care cost / overhead costs (USD).**   The overhead costs that were shared by the patients included human resource costs (79.33USD), SD = 4.63) and maintenance and utilities (2.04USD), SD = 0.27) giving rise to a total of 81.37USD.

These costs are summarised in Table 2 below.

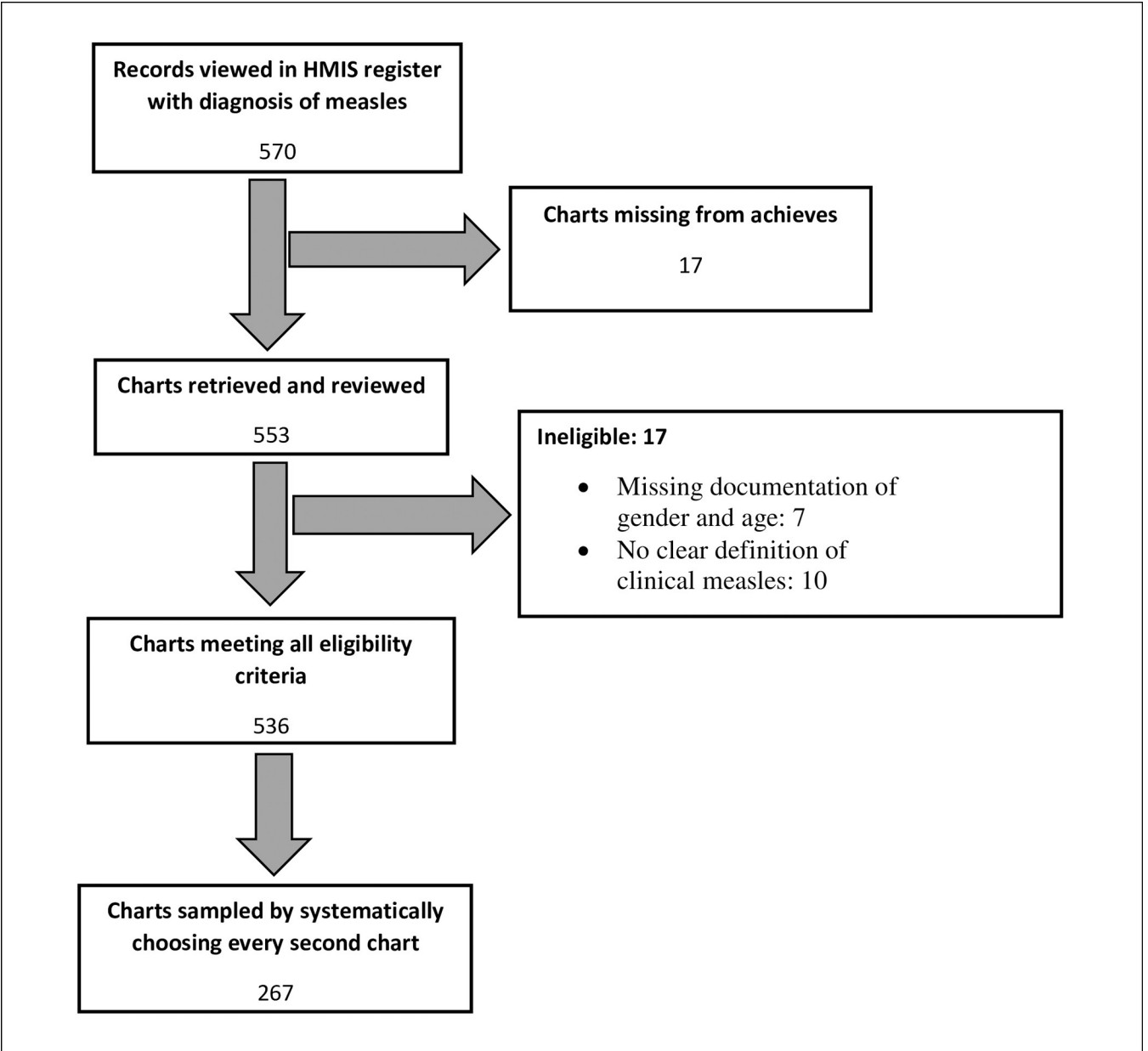

**Fig 1. Selection of charts included in analysis.**

**Comparison of direct costs incurred by patient characteristics.** Male patients incurred higher average costs (36.08USD, SD = 16.04) than their female counterparts (30.24USD, SD = 24.89), (F = 6.126, P = 0.011). Patients that had comorbidities/long standing illness recorded during admission period incurred higher average costs (45.00USD, SD = 36.49) than those who had no comorbidities (32.02USD, SD = 29.67), (F = 2.163, P = 0.04).

Patients who stayed for over 15 days in the hospital incurred higher average costs (127.08, SD = 56.36) than those who stayed between 8–15 days (57.07, SD = 25.68), and those who stayed for less than 8 days (22.81, SD = 15.08), (F = 170.262, P = 0.0001).

**Table 1. The demographic and clinical characteristics of 267 patients with clinical measles admitted to Mulago Hospital from 1st January 2018 to 31st December 2018.**

| Variable | Category | n (%) |
|---|---|---|
| **Gender** | Male | 132 (49%) |
| | Female | 135 (51.%) |
| **Age** | Age in years, Median (IQR) | 1.0 (0.75–2.0) |
| | < 1 | 169 (63%) |
| | 1–5 | 83 (31%) |
| | >5 | 15 (6%) |
| **Duration of hospital stay** | Median (IQR) | 4.0 (3.0–7.0) |
| | 0–7 | 207 (77%) |
| | 8–15 | 50 (19%) |
| | >15 | 10 (4%) |
| **Status at discharge** | Alive | 237 (89%) |
| | Dead | 30 (11%) |
| **Immunization status** | Yes* | 63 (24%) |
| | No¥ | 79 (30%) |
| | Unknown# | 69 (26%) |
| | Not due** | 56 (21%) |
| **Co-morbidities at admission** | Yes | 23 (9%) |
| | No | 244 (91%) |

\* Received the measles vaccine

¥ Did not receive the measles vaccine

# Not recorded

\*\* Less than 9months old.

At the time of discharge, the patients who died had incurred higher average cost (47.13, SD = 34.28) than those discharged alive (31.36, SD = 19.04), (F = 4.49, P = 0.03). These findings are summarized in the **Table 3** below.

## Discussion

We carried out a retrospective cohort study to describe the immediate outcomes of children admitted with clinical measles and to estimate the cost of managing a severe measles case in Mulago National Referral Hospital, Uganda.

### Immediate treatment outcomes

Complicated measles runs a severe course and can lead to death. In this study, mortality was higher than that found in two studies done in hospitals in Pakistan (5.4% and 3.4%) [14,15]. These hospitals may not be entirely similar to Mulago hospital, but are both national referrals and teaching facilities, offering quite similar services to those offered at Mulago Hospital. Mortality was found highest among the children who were not immunised, possibly because they developed severe complications. Measles related mortality is primarily due to an increased susceptibility to secondary bacterial and viral infections, resulting from direct mucosal damage by measles infection and measles induced immune suppression [16]. While pneumonia was the most prevalent complication, acute watery diarrhoea (gastroenteritis) was also common. These complications were similarly found to be the most prevalent in a study done in Pakistan with pneumonia and diarrhoea reported in 39.7% and 38.2% of children respectively [17].

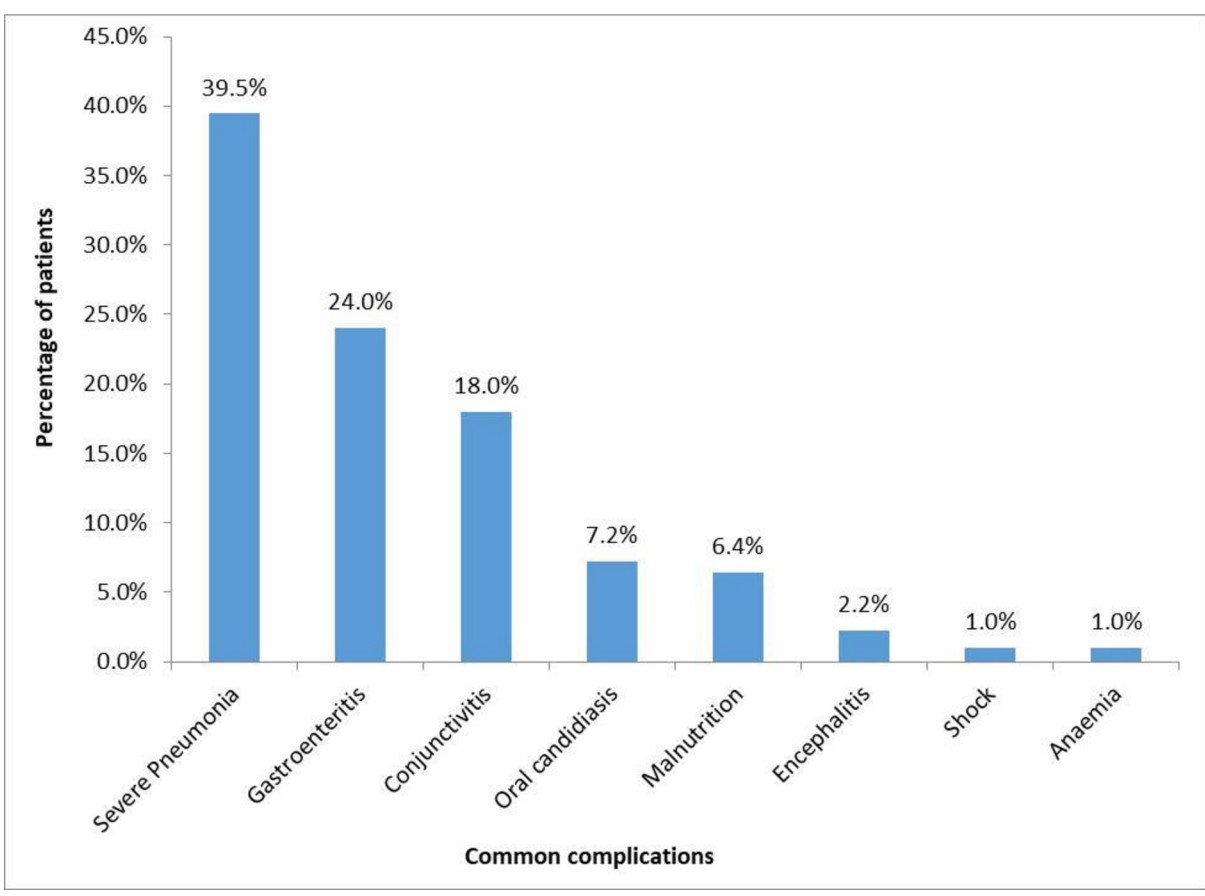

**Fig 2. Common complications.**

Pneumonia, just like measles in children, is largely preventable through vaccination. In children who are not immunised, it can lead to severe disease which may result in death. This study showed that pneumonia was the commonest complication among children who died, having been diagnosed in more than a quarter of these; and since majority (44.0%) were not immunised, it is possible that they had also missed the rest of the vaccinations including those for pneumococcal pneumonia and influenza. This was similarly demonstrated in a study done in Queen Elizabeth Central Hospital in Malawi where pneumonia was the greatest contributor to mortality [18]. However it is in contrast to findings from a study done in Ayub teaching hospital in Pakistan where encephalitis was the leading cause of death [17].

The median length of hospital stay was 4.0 days with about three quarters of children staying less than seven days. This was similarly found in a study done in Pakistan about the clinical outcome of hospitalised measles patients where the mean hospital stay was 3.8 days [17].

### Estimated cost of managing a child with measles

In management of children hospitalised due to measles, both healthcare costs and non-health care costs are incurred. However, this study looked at only the costs borne to the health care delivery system. The study found that the estimated direct cost borne by the health care system (laboratory test, radiology and treatment plus supplies cost) was on average 122,500 Ugandan shillings which is approximately 33.13 US dollars (1 US Dollar = 3697.92 UGX) [19], 66.3% of which goes to treatment and supplies.

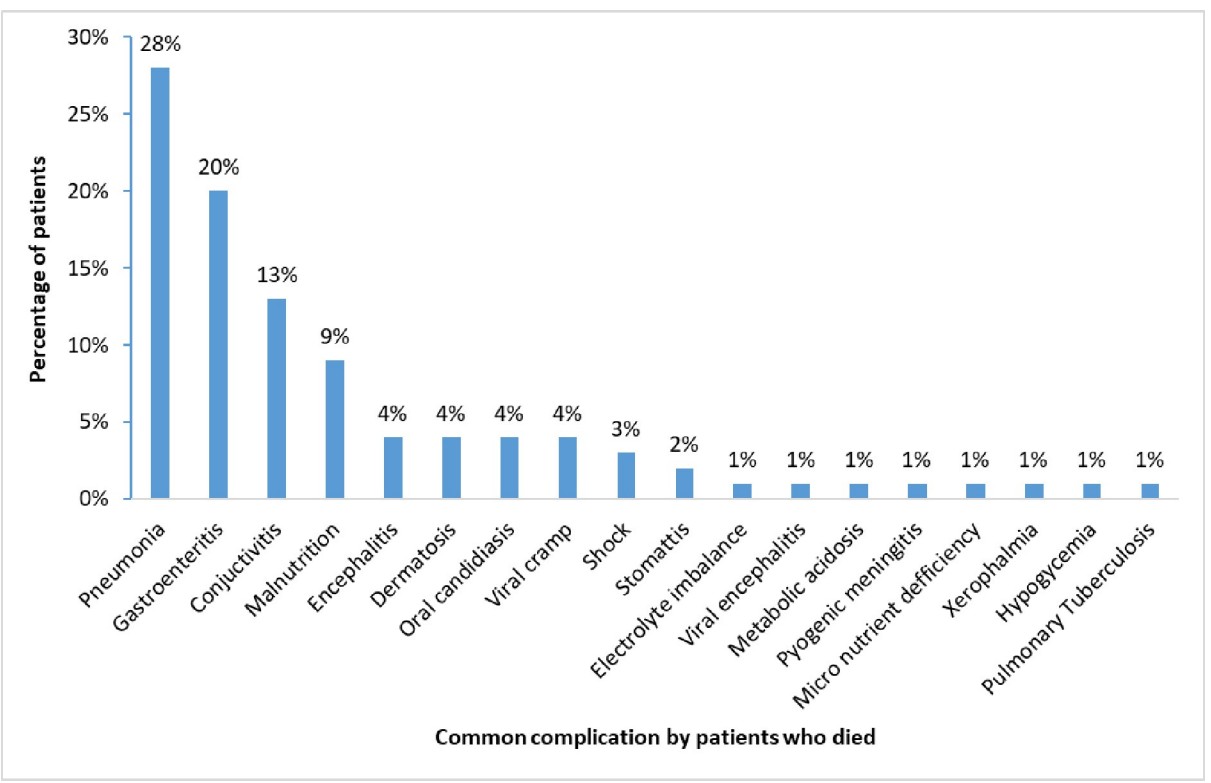

**Fig 3. Common complications by patients who died.**

The average overhead cost was 81.37 US dollars, most of which goes to human resource (salaries) to care for these patients. Thus, the total estimated direct cost of treating a measles case was substantially high (114.50 USD). This is higher than the cost determined by Gatien De Broucker and colleagues in another study conducted in Uganda [20]. Our costs were higher possibly because the study was conducted in a National Referral Hospital where severe cases of measles are referred.

Such costs are far beyond the cost required to immunise a child against measles which the World Health Organisation estimates to be less than a dollar [9,21]. Whereas there was no difference in hospital costs of the immunised and unimmunised children, complications are present in majority of fatal measles cases and these cases carry a higher cost to the health care delivery system. This provides some ground to base on to promote vaccination.

**Table 2. The average direct health care and non-health care (overhead) costs for managing measles per patient.**

| Health care costs | Item | Average cost (USD) | Standard deviation | Range (USD) |
|---|---|---|---|---|
| | Blood test | 11.73 | 12.10 | 1.35–59.49 |
| | Radiology | 3.48 | 4.83 | 1.35–43.27 |
| | Treatment | 21.98 | 22.77 | 2.57–211.22 |
| | **Total cost** | **33.13** | **30.45** | 4.80–216.64 |
| Non-health care costs | Human resource | 79.33 | 4.63 | 1.35–17.56 |
| | Maintenance, utilities, | 2.04 | 0.27 | 0.02–8.18 |
| | **Total cost** | **81.37** | **14.09** | |

**Table 3. Comparison of direct costs (USD) incurred by patient characteristics.**

| Variable | Category | Average cost (SD) | F-Statistic | P-Value |
|---|---|---|---|---|
| **Gender** | Male | 36.08(16.21) | 6.126 | 0.011 |
| | Female | 30.24(24.89) | | |
| Age (Years) | <1 | 34.76 (29.29) | 2.206 | 0.112 |
| | 2–5 | 28.04 (28.14) | | |
| | Over 5 | 42.98 (30.02) | | |
| **Immunization status** | Yes | 36.49(29.21) | 0.738 | 0.530 |
| | No | 35.40 (27.62) | | |
| | Unknown | 30.39 (24.71) | | |
| | Not due | 30.04 (22.10) | | |
| **Comorbidities** | Yes | 45.00(36.49) | 2.163 | 0.04 |
| | No | 36.02 (29.67) | | |
| **Hospital days** | 0–7 | 22.81 (15.08) | 170.262 | 0.0001 |
| | 8–15 | 57.07 (25.68) | | |
| | >15 | 127.08 (56.36) | | |
| **Status at discharge** | Dead | 47.13 (34.28) | 4.49 | 0.03 |
| | Alive | 31.36 (19.04) | | |

In this study, higher costs of treating a measles case were found among the male gender, those with co-morbidities, those that were admitted longest (> 15 days), and those that died during admission. Not surprisingly, patients that had co-morbidities during admission incurred higher average costs (44.99 USD) than those who had no co-morbidities. The possible explanation for this is that those with co-morbidities had to be managed for both measles in addition to other illnesses. It is also possible that those with co-morbidities had severe measles infection due to higher immune suppression, both by the measles and the co-morbidity. Management of such patients requires more investigations and perhaps more expensive treatment, which in turn increases the cost of health care delivery.

Patients who also stayed longer in hospital (> 15 days) incurred higher average costs than those who stayed for less days. These children were possibly too sick and required more aggressive and/or prolonged treatment and investigations which increased the direct cost of health care delivery. At the time of discharge, those who died incurred more average direct costs (47.13 USD) than those who were alive despite the fact that they spent less days. This is possibly because those who died were severely sick and required more expensive treatment and more investigations to be done than those who were discharged alive. Managing a male patient was found to be costlier than a female one, possibly because as demonstrated, males were admitted longer and were more likely to die than females.

To our knowledge, this is the first study that has estimated the cost of managing a child with clinical measles in a National Referral Hospital in Uganda.

This study had limitations. It was done in a public hospital and findings may not be similar in private facilities. It is more likely that costs in private facilities are even higher since these are established to make profit as was found by Gatien De Broucker et al [20].

Since this study was retrospective, it was only possible to enumerate costs to the health care delivery system but not those incurred by the family or society as these would require real time interaction with the family. It would have been useful to document costs incurred by the family to support messages to promote vaccination among parents/care givers. Measles serology was not done to confirm the diagnosis in the hospital. There is therefore a possibility that some diagnoses of measles were actually other viral exanthems like rubella, scarlet fever, varicella

and roseola. It was also hard to verify whether all the treatments/investigations documented in the charts were received by the patients. We instead assumed that they were received since they would ideally be required for management of these cases. However, in spite of these limitations, we believe our findings are valid and pertinent, in efforts to promote immunisation against measles.

## Conclusion

Complications are common in majority of fatal measles cases and these carry a high cost to the healthcare system. More research is necessary in this setting to quantify the economic burden of measles to families and society. We recommend that efforts that promote uptake of the cheaply available measles vaccine are implemented to reduce on incidence of severe disease that requires hospitalisation.

## Supporting information

**S1 File. Data abstraction form.**
(DOCX)

**S1 Table. Immunisation status of the 185 children with complications.**
(DOCX)

**S2 Table. Demograhic characteristics of participants who died.**
(DOCX)

**S3 Table. Characteristics of participants who stayed longer than 7 days.**
(DOCX)

## Acknowledgments

We acknowledge the department of Paediatrics Makerere University, the research assistants, Bbosa Juliet, Omega Jotham, and the participants whose information was used.

## Author Contributions

**Conceptualization:** Barbara Namugga.

**Data curation:** Barbara Namugga, Leonard Manirakiza.

**Formal analysis:** Barbara Namugga, Leonard Manirakiza.

**Funding acquisition:** Cecily Banura.

**Investigation:** Barbara Namugga.

**Methodology:** Barbara Namugga.

**Supervision:** Ombeva Malande, Jonathan Kitonsa, Cecily Banura, Ezekiel Mupere.

**Validation:** Barbara Namugga, Leonard Manirakiza.

**Visualization:** Barbara Namugga.

**Writing – original draft:** Barbara Namugga.

**Writing – review & editing:** Ombeva Malande, Jonathan Kitonsa, Leonard Manirakiza, Cecily Banura, Ezekiel Mupere.

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
