## [Decision Letter · Decision Letter 0]

23 Feb 2023

PGPH-D-22-02083

The immediate treatment outcomes and cost estimate for managing clinical measles in children admitted at Mulago Hospital: a retrospective cohort study.

Dear Dr. Namugga,

Thank you for submitting your manuscript to PLOS Global Public Health. After careful consideration, we feel that it has merit but does not fully meet PLOS Global Public Health’s publication criteria as it currently stands. Therefore, we invite you to submit a revised version of the manuscript that addresses the points raised during the review process.

We look forward to receiving your revised manuscript.

Kind regards,

Abram L. Wagner, PhD, MPH

Academic Editor

Journal Requirements:

2. Please remove any figures embedded in your manuscript file. Please also ensure that all files are under our size limit of 10MB.

Additional Editor Comments (if provided):

Reviewers' comments:

Reviewer's Responses to Questions

**Comments to the Author**

1. Does this manuscript meet PLOS Global Public Health’s publication criteria? Is the manuscript technically sound, and do the data support the conclusions? The manuscript must describe methodologically and ethically rigorous research with conclusions that are appropriately drawn based on the data presented.

Reviewer #1: Partly

Reviewer #2: Partly

Reviewer #3: Partly

2. Has the statistical analysis been performed appropriately and rigorously?

Reviewer #1: Yes

Reviewer #2: I don't know

Reviewer #3: Yes

3. Have the authors made all data underlying the findings in their manuscript fully available (please refer to the Data Availability Statement at the start of the manuscript PDF file)?

Reviewer #1: Yes

Reviewer #2: No

Reviewer #3: No

4. Is the manuscript presented in an intelligible fashion and written in standard English?

Reviewer #1: Yes

Reviewer #2: Yes

Reviewer #3: Yes

5. Review Comments to the Author

Reviewer #1: This a relevant and timely topic as the world is experiencing many measles outbreaks in the COVID era. Evaluating the cost of treatment for preventable diseases provides a solid evidence-based argument to strengthen preventive strategies. However, for a more robust methodologic approach and better clarity, we offer those comments:

Objective

- For more clarity, please precise the study period in the objective. And specify that the "average cost" was determined.

Method

- Why is the study defined as a single cohort study? The method and data presented are more in favor of a transversal study, not a longitudinal approach. If there is additional information that can support the Cohort aspect, please share.

- The study procedures (Section Method) stated: “Market prices were used for those items that were not in the price list.” As it is a retrospective study, we wonder if there were any fluctuations between the current prices (prices during the data collection) and those in 2018. If yes, which one was used, and how did the authors avoid bias considering this outcome? The clarity of the study will benefit from that precision.

- It would help the readers’ comprehension if treatments cost was defined. What that variable included? (IV, drugs, equipment utilization…)? Additionally, it is better to keep the same classification of direct costs in both methods and results. In the Methods, the direct costs are defined as direct health care costs (drugs, supplies, and investigations)

Results

- In the results section, “Direct health care costs were divided into three categories, i.e., blood tests, radiology, and treatment costs.” Blood tests and radiology can be labeled as investigations; drugs can be considered treatment. Are supplies included in the treatment group? The reader should not have to guess.

Discussion

- When considering the study objectives and the results presented, we realize that this study explores not only the direct costs as stated in the discussion. I would suggest removing the word "direct."

General comment

- It is unclear if the authors used only the clinical admission diagnosis of measles to select the study sample. If yes, it could have been interesting to know the proportion of confirmed cases and compare the costs between confirmed cases and suspect cases.

Reviewer #2: Overall:

The paper is setting out to identify the direct and indirect clinical costs of managing measles cases, which it does ($114.5 USD ave cost estimate in reference to $0.86 cost of measles vaccination). From here the authors go forward to say that these high costs of care are “strong ground to base on to promote vaccination,” however the data presented in this paper shows that immunization status has no statistical difference on direct hospital costs incurred ($36.49 immunized patient, $35.40 non-immunized patient). A more accurate conclusion should state that complications/comorbidities are present in a majority of fatal measles cases and comorbid cases carry a higher cost to the healthcare system.

If they want to move forward with their argument that immunization reduces case management costs, then it needs to be demonstrated that non-immunized patients had higher comorbidities (not addressed in complicated patients n=185), non-immunized patients were admitted longest (briefly addressed in S2), and that non-immunized patients were more likely to die during admission (addressed).

If this is a paper establishing costs of managing measles cases, then the paper has initial merit but needs expanded information on how direct and indirect costs were determined.

If this paper wants to make claims that immunization decreases costs of managing measles cases, then further analysis must be included.

Comments on Introduction:

1. Edit hard immunity to herd immunity.

Comments on Study Procedure:

2. Participant charts were identified from hospital diagnosed measles cases but some charts did not meet criterion for clinical measles and therefore were excluded. Although WHO Guidelines are referenced, it would be good for the reader to have an idea of the study’s inclusion/exclusion criteria in more detail.

3. Age of inclusion is mentioned in the abstract, it is never mentioned in the Study Procedure section.

4. This is a study to establish the costs of managing measles cases, but there are no methods listed describing how non health care costs were actually determined. For security and staff, were the total number of staff salaries per day determined then divided by the daily number of patients in hospital? Does measles case coverage necessitate a higher staff per patient ratio? For utilities, is this a hospital that relies on generator electricity with high fuel costs or is there provision of consistent electricity? If this is claiming to be a health economics paper, then please provide additional detail and methods of how costs are determined.

5. Data Abstraction Form (DAFs) could be shared in supplemental materials.

Results:

6. No information is give about the Immunization status Not Due. A quick Google search shows me that US CDC guidelines for measles immunizations are 12-15 months, WHO guidelines are 9 months. This is very relevant information seeing that age range for participants is 0-12 years, the median age of participants is 1.0 year, a majority of patients are aged less than 1 year. Please provide justification, in terms of health outcomes, why patients in the Not Due category do not fall in the non immunized category.

7. Please provide information/justification why Immunization status Unknown was not considered as a case with missing documentation and therefore subject to exclusion from the study.

8. The text states that Table 1 displays 30 deaths and 237 discharges, but Table 1 does not contain this information.

9. Standardize the description/name of your n=185 Severe Illness group. Table 1 lists Co-morbidities at admission (n=23) and Figure 1 describes Common Complications during period of admission (n=185). Later in Comparison of Direct Costs Incurred by Patient Characteristics it states patients with co-morbidities incurred higher costs than those with no comorbidities and you are talking about your n=185 group but it could be misconstrued as your n=23 group. The Common Complications figures are never referred to as comorbidities but the n=23 group is referred to as comorbidities.

10. Regarding Common Complications, you have information of patients who died (n=30), but it would also be helpful regarding Common Complications (n=185) to have information of immunization status.

11. For Duration of Hospital Stay and Characteristics Among the 60 Patients Who Stayed Longer Than 7 days, you say 207 (77%) stayed <7 days and 60 (13%) stayed >7 days, however 77%+13% does not equal 100%. Later in Table 3 you present information on patient/s who stay >15 days but this group is not mentioned in the Characteristics text section. Please give the n of patients who stay >15 days. Patients who stayed over 15 days in the hospital incurred a higher average cost; is that just a cumulative effect of the number of days, or is their daily rate higher? And of the participants who stay >7 days, a majority were less than one year; this brings up topics around the age of immunization initiation which is never brought up in the discussion.

12. Table 2 lists “Treatment” costs. What are Treatment costs? Drugs? The Study Procedure section says direct health care costs are drugs, supplies, and investigations, but Table 2 lists direct health care costs as Blood Test, Radiology, and Treatment. Please clarify terminology and how costs were estimated and categorized.

13. Table 3 displays that immunized and non immunized patients cost the hospital the same amount. Supplemental Table 1 does establish that 13/30 of the high-cost-Dead at Discharge individuals are non immunized, but no analysis is done of the 185 high-cost-Comorbidity individuals regarding immunization status.

Discussion Comments:

14. Many comparisons with Pakistan but no context to similarity of hospitals, health systems, etc.

15. Do not refer to participants as ‘babies’, inclusion criteria was 0-12 years.

16. It is stated that immunization costs are less than a dollar and hospital treatment costs are high, “this is strong ground to base on to promote vaccination,” however the data in this study showed no difference in hospital spending when comparing immunized vs. non immunized.

17. “Since vaccines are available at health facilities, it is clear that the most important challenge is related to utilization/demand.” There is no basis for this statement, no regard or discussion of societal and economic barriers to health facility access.

Conclusion:

18. “Efforts are needed to promote immunization against measles and related mortality and thus reduce costs to the healthcare delivery system.” The data presented in this paper does not support this conclusion. A more accurate conclusion states that complications/comorbidities are present in a majority of fatal measles cases and comorbid cases carry a higher cost to the healthcare system.

Reviewer #3: This manuscript describes the burden of measles in hospitalized cases in an Ugandan national hospital. The costs of medical care and hospitalization are calculated and described. The authors analyze the factors associated with increased hospital cost. Overall, this paper is structured well. The English writing is adequate. The data combine clinical information and cost, thus providing useful information for their country’s national program to improve vaccination coverage and follow up.

Minor comments

There are both UK and American English spellings in the paper that need to be resolved. Be careful to make the discussion and conclusions based on the data and not overly speculative.

Abstract

No comments here, but some changes may be needed that are related to comments in the body of the manuscript.

Introduction

Overall: Suggest to include a map to show the percentages of measles vaccination and where the outbreaks are.

“Besides morbidity and mortality, costs related to treating a child with measles pose a big burden to the family and the already financially stretched health care delivery system, and are increased when the measles is complicated.” It would be helpful to give some details of what defines measles complications.

Methods

Study design and setting: please describe the national guidelines for measles vaccine in your setting.

“Those with incomplete data like undocumented inpatient number, age,and sex were excluded.” List all inclusion and exclusion critiera. A trial diagram should be presented and I suggest that the authors use the STROBE checklist to confirm their work.

“The first chart was selected using simple random sampling.” Give the details of the random sampling rather than using the word ‘simple’.

“Information abstracted from the patient’s charts included: Demographics (age, gender)….” If you mean male or female, then ‘sex’ will be more precise than ‘gender’, as gender has many categories.

“These were checked against the patient’s record daily for completeness and accuracy.” I do not understand why the DAF's were checked against the patient record every day. Is this not a retrospective study?

“School of Medicine Research and Ethics Committee (SOMREC) and The Mulago Hospital Institution Review Board.” Provide the ethical approval number of the EC.

Results

“A total of 570 patients were admitted between 1st January and 31st December 2018

with a clinical diagnosis of Measles and we managed to retrieve 553…” Measles should not be capitalized. '...we managed to retrieve....' could be changed to something more objective such as ' ..... and 553 charts were retrieved...'

Table 1: Definitions of Immunization status need to be more clear. Does 'yes' mean up to date or had at least one vaccine? Does 'no' mean never before or due? Does 'Not due' mean up to date? Do immunizations include any vaccines or only measles vaccine? It would be interesting to know how many were too young to have received measles vaccine, how many were eligible but did not receive measles vaccine. This is important because the infants who were not eligible for measles vaccine (due to their age) should not be included in the cost analysis as measles vaccine access is not the issue. This must be specified in the study methods.

I also suggest to present whole percentages as the addition of the decimal point does not add any more precision.

“Out of these, 17 (57%) were male. 21 (70%) of these were aged between 0-1 years and 13 (44%) were not immunized.” Of the ones not immunized, how many were were <12 months (or 9 months) and not yet eligible for vaccination?

Figure 2: Please clean up the x-axis labels. There should be no underscores and PTB needs to be defined.

“The median length of hospital stay among children admitted with measles was 4.0 (IQR

3.0-7.0) days.” This result is already mentioned above. Please report it only once.

“Of these, 46 (77 %), were less than one year and 36….” Of which one - the ones that stayed less than or more than 7 days?

Table 2: Clean the Item names (ie, Maintenance)

“Male patients incurred higher average costs (36.08USD, SD=35.09) than their female

counterparts (30.24USD, SD=24.89), (F=6.126, P=0.011).” There may be correlating variables here. For example, were the patients with co-morbidities more likely to have complications? Were males more likely to have co-morbidities? Hospital duration should be included here. Please perform either a multivariable ANOVA or linear regression that includes all of the potentially related variables in the model. This also would need to be described in the statistical section.

“Patients who stayed for over 15 days in the hospital incurred higher average costs….than those who stayed between 8-15 days….” The reader does not know how many persons stayed >15 days. This should be presented above. I would suggest using similar groupings and the same definition as what were presented above (more than 7 days).

Table 3: Double check all the numbers in this table. All the standard deviation values seem too high and definitely should not greater than the mean. The number of deaths does not match the text, where around 10% of the cases had died (not 45% as is stated in this table). Hospital stay and complications may be collinear - this probably should be addressed in the results. Once you have shown that they are collinear, perhaps only one of them (either hospital stay or complications) needs to be presented.

Discussion

“Mortality was found highest among the children who were not immunised, possibly

because they developed severe complications.” With your data, you can run this analysis using a multivariable ANOVA or logistic regression model (death yes or no) which includes the related variables (sex, co-morbidities, hospital stay/complications, immunization status, etc).

“……..it is possible that they had also missed the rest of the vaccinations including those for

pneumococcal pneumonia and influenza.” Please be careful what conclusions are made here. From the results, I have not been able to determine which infants received a vaccine (which should include pneumococcal pneumonia if they are <6 months old), but not yet measles (because they were not yet eligible by age). And the data does not show which cases that missed the measles vaccine visit also missed all of the previous vaccine visits. It is important to differentiate those children who have missed a measles vaccine opportunity versus those who are not yet eligible (and should not be included in the cost analysis).

“…..was on average 122,500

Ugandan shillings which is approximately 33.13 US dollars (1 US Dollar=3697.92 UGX) [17], much of which goes to treatment and supplies.” How much goes to treatment and supplies? (A percentage would be interesting here).

“Such costs are far beyond the cost required to immunise a child against measles which the

World Health Organisation estimates to be less than a dollar.” How much is a measles vaccine in your setting?

“The possible explanation for this is that those with co-morbidities had to be managed for both measles in addition to other co-morbidities.” Is it more expensive because they had higher risks for complications and stayed in the hospital longer? This will need to be addressed in the analysis and results.

“Patients who also stayed longer in hospital (> 15 days) incurred higher average costs than those who stayed for less days.” Please refer to the comment previous. Complications, co-morbidities and hospital stay are likely related. This needs to be addressed in the analysis.

“There is therefore a possibility that some diagnoses of measles were

actually other viral exanthems like rubella, scarlet fever, varicella and roseola.” Suggest specifically stating that measles serology was not performed.

“We instead assumed that they were received since they would ideally be required

for management of these cases.” This is an interesting point especially to readers who live in cultures where this occurs less often or rarely. It might be worthwhile adding it to your statistical section or study methods – specifically stating that you are assuming all costs incurred were received by the patient.

“The estimated average direct health care cost of treating a measles case in the National Referral Hospital was considerably high.” I suggest adding a 'cost savings' value. For example, of the cases in this publication, if 90% (since a single dose of MMR vaccine should result in 90% of the population to be immune) hospitalizations (xx numbers of cases) were prevented with measles vaccination, then xxx dollars (total cost minus the vaccine cost) would have been saved by the health system.

Acknowledgements

“…and the participants whose charts we used.” Suggest changing the word ‘charts’ to ‘information’.

6. PLOS authors have the option to publish the peer review history of their article (what does this mean?). If published, this will include your full peer review and any attached files.

**Do you want your identity to be public for this peer review?** For information about this choice, including consent withdrawal, please see our Privacy Policy.

Reviewer #1: No

Reviewer #2: No

Reviewer #3: No

---

## [Editor Report · Decision Letter 1]

31 May 2023

PGPH-D-22-02083R1

The immediate treatment outcomes and cost estimate for managing clinical measles in children admitted at Mulago Hospital: a retrospective cohort study.

Dear Dr. Namugga,

Thank you for submitting your manuscript to PLOS Global Public Health. After careful consideration, we feel that it has merit but does not fully meet PLOS Global Public Health’s publication criteria as it currently stands. Therefore, we invite you to submit a revised version of the manuscript that addresses the points raised during the review process.

We look forward to receiving your revised manuscript.

Kind regards,

Abram L. Wagner, PhD, MPH

Academic Editor

Journal Requirements:

Additional Editor Comments (if provided):

We appreciate your edits to the manuscript. I have just a few minor issues to address.

Could you provide more detail about the sample size? You mention sample frame and that you only needed half of that, but could you explain why? (for instance, something like: "to estimate [either a relationship or a statistic] we would need a sample size of XXXX with an alpha of 0.05 and a power of 80%")

Could you edit the figures? In essence, try to make them have higher quality in terms of DOI and be conscientious of how you use screenshots. For one, remove the spell check so there aren't wiggly lines underneath the text. I believe you are using Microsoft Office. Some people have found ways of exporting higher quality images (for instance, see https://learn.microsoft.com/en-us/office/troubleshoot/powerpoint/change-export-slide-resolution) using PowerPoint. If you have another method, that possibly could work.
---

## [Editor Report · Decision Letter 2]

23 Jun 2023

The immediate treatment outcomes and cost estimate for managing clinical measles in children admitted at Mulago Hospital: a retrospective cohort study.

PGPH-D-22-02083R2

Dear MD Namugga,

We are pleased to inform you that your manuscript 'The immediate treatment outcomes and cost estimate for managing clinical measles in children admitted at Mulago Hospital: a retrospective cohort study.' has been provisionally accepted for publication in PLOS Global Public Health.

Best regards,

Abram L. Wagner, PhD, MPH

Academic Editor

Thank you for your work responding to prior critiques. I recommend the following changes to the abstract, but this is not required prior to publication:

The end of abstract intro is "Whereas the cost of measles vaccination is

known, the cost of treating measles is unknown. Knowledge of this can inform policy

and planning for healthcare services."

I would instead switch this to the aims "This study aims to ...."

For beginning of abstract results, you write "Out of 536 patient charts, 267 were chosen, 51% were for females and the

median age was 1.0 (IQR 0.75-2) years."

I would simplify this to: "Among 267 reviewed patient charts, the median age was 1.0 (IQR 0.75-2) years."